# Protein Concentrations in Stored Pooled Platelet Concentrates Treated with Pathogen Inactivation by Amotosalen Plus Ultraviolet a Illumination

**DOI:** 10.3390/pathogens11030350

**Published:** 2022-03-14

**Authors:** Niels Arni Arnason, Freyr Johannsson, Ragna Landrö, Björn Hardarsson, Sveinn Gudmundsson, Aina-Mari Lian, Janne Reseland, Ottar Rolfsson, Olafur E. Sigurjonsson

**Affiliations:** 1The Blood Bank, Landspitali-The National University Hospital of Iceland, 105 Reykjavik, Iceland; nielsa@landspitali.is (N.A.A.); ragnal@landspitali.is (R.L.); bjornh@landspitali.is (B.H.); sveinn@landspitali.is (S.G.); 2School of Engineering, Reykjavik University, 105 Reykjavik, Iceland; 3Department of Medicine, University of Iceland, 105 Reykjavik, Iceland; johannsson.freyr@mayo.edu (F.J.); ottarr@hi.is (O.R.); 4Institute of Clinical Dentistry, Faculty of Dentistry, University of Oslo, 0317 Oslo, Norway; a.m.lian@odont.uio.no (A.-M.L.); j.e.reseland@odont.uio.no (J.R.)

**Keywords:** platelet concentrates, pathogen inactivation, proteins, cytokines, chemokines, growth factors, platelet storage lesion

## Abstract

Platelet granules contain a diverse group of proteins. Upon activation and during storage, platelets release a number of proteins into the circulation or supernatant of stored platelet concentrate (PC). The aim of this work was to investigate the effect of pathogen inactivation (PI) on a selection of proteins released in stored platelets. Materials and Methods: PCs in platelet additive solution (PAS) were produced from whole blood donations using the buffy coat (BC) method. PCs in the treatment arm were pathogen inactivated with amotosalen and UVA, while PCs in the second arm were used as an untreated platelet control. Concentrations of 36 proteins were monitored in the PCs during storage. Results: The majority of proteins increased in concentration over the storage period. In addition, 10 of the 29 proteins that showed change had significantly different concentrations between the PI treatment and the control at one or more timepoints. A subset of six proteins displayed a PI-related drop in concentration. Conclusions: PI has limited effect on protein concentration stored PC supernatant. The protein’s changes related to PI treatment with elevated concentration implicate accelerated Platelet storage lesion (PSL); in contrast, there are potential novel benefits to PI related decrease in protein concentration that need further investigation.

## 1. Introduction

In circulation, platelets modulate an immune response [1,2,3] and maintain the integrity of the endothelial cell lining of the vascular system as part of the coagulation process in hemostasis [4,5]. Stored platelets play a key role in transfusion medicine. They are used for actively bleeding patients, though most platelet concentrate (PC) units are issued as prophylaxis for patients undergoing myeloablative therapy (with resulting low platelet counts) [6,7]. 

Modern transfusion medicine demands safe PC products with increased availability. Countries with ageing populations, increased hospitalization and limited donations face an ever more challenging task in meeting harvesting and storage demands [6,7,8,9,10,11]. Unfortunately, PC shelf life is normally limited to 5 days (using standard room temperature storage with gentle agitation) [12,13,14] due, in part, to higher risk of bacterial growth in PCs than in blood components stored at cold temperatures [15,16]. This risk has been reduced by improvements in aseptic procedures during collection and processing, donor selection protocols, and bacterial screening. However, bacterial contamination of stored PCs and transfusion-transmitted bacterial infections persist, and septic transfusion reactions still occur [14,16,17,18,19]. A retrospective analysis of PC transfusions indicates that the rate of septic transfusion reactions is likely underestimated [20].

One approach to improving PC shelf life is to apply pathogen inactivation (PI) to PCs using ultraviolet (UV) light and photoactive chemicals. Among the available PI, the amotosalen plus UVA light system represents the most studied and widely used method [21]. Amotosalen intercalates into nucleic acids, and, upon UVA light treatment, forms permanent adducts that crosslink at high frequency, inhibiting strand separation and proliferation of pathogens and residual white blood cells. The amotosalen-UVA method is effective against a broad spectrum of viral, bacterial and protozoan pathogens and, potentially, against unrecognized blood-borne pathogens, as well as being effective in preventing Graft vs. Host disease [22,23,24].The extended shelf life in turn increases PC availability, while eliminating the need for a quarantine period for bacterial screening [25,26,27,28,29,30,31]. Even with all current safety measures including PI, PC shelf life is commonly limited to 7 days at the most. Platelet storage lesion (PSL) is the second limiting factor of PC shelf life. Stored platelets retain their metabolic activity, which acidifies the storage media and leads to deterioration of platelet quality during storage [32,33,34]. PSL can be monitored based on the number of activated or damaged platelets in the PC [35]. During storage a variety of factors are released into the storage media via de-granulation of activated platelets. Platelets and their granules contain various proteins at high concentrations that act as biological response modifiers (BRMs) [3]. Numerous factors, including donation, post-collection manipulation, and storage, affect platelet quality [36,37,38,39,40,41], and it is well documented that the preparation and storage of platelets can induce platelet activation and release of BRMs [42,43,44]. Platelet lysis due to storage and processing, as well as apoptosis and platelet senescence, can also contribute to BRM release into the platelet storage media [45,46,47,48,49,50]. Recently, the role of BRM released from platelets in modulating immune response has become more evident [2,3,51,52]. For specific BRMs, there is a positive correlation between storage time and extracellular concentration. sCD40L [42] and sP-selectin [53] are frequently used as indicators of PSL [35]. CD40L has clinical relevance and is implicated in various transfusion-related adverse events (TRAEs), the most serious being transfusion-related acute lung injury [54,55,56]. Other platelet related BRM include: RANTES, a pro inflammatory chemokine implicated in none hemolytic and allergic TRAE [57,58]; IL-27 and sOX40L have implications in febrile nonhemolytic transfusion reactions [59]; IL-13 and MIP-1α levels seem to be reliable predictors of TRAEs [60]; VEGF, which has implications in atherosclerosis, promotes tumor metastasis, and negatively effects cancer treatment [61,62]; PF4, an inhibitor of cell proliferation and culprit in heparin induced thrombocytopenia (HIT) [63,64]; and PDGF, TGF-β, EGF, IL-8, and IL-7, which all have various roles in inflammation, cell recruitment, and phagocytosis [65]. Post-transfusion recovery and survival have been correlated with PSL [66,67].

In this study, we analyzed the potential impact of PI treatment on a selection of proteins during 7 days of storage under standard blood banking conditions. We compared an amotosalen-UVA-treated PC in platelet additive solution (PAS) with a non-pathogen inactivated PC in PAS (control).

## 2. Results

### Protein Accumulation in PCs during Storage

With a relatively large data set of 4 sampling days and 36 proteins and 8 biological replicates (n), we used PCA plots to calculate and display the variation in our data set. Protein concentrations during day 2–7 do not display treatment related separation in the first principal component (PC 1) accounting for overall 61% of the variation. The second principal component (PC 2) revealed some visible separation between C-PAS and PI-PAS accounting for 12.2% of the total variation (Figure 1A). When baseline data was included and individual sampling days were plotted, differences in protein concentrations during storage are more visible in PC 1, accounting for 61.6% of the total variation (Figure 1B). The main variation in the data set, PC 1 is driven by changes in protein concentration relating to storage time. The second largest variation, PC 2 is treatment related. 

To further analyze the impact of PI on protein concentrations, repeated-measures ANOVA was used to compare the results from C-PAS and PI-PAS on Days 2, 4, and 7. Thirty proteins were significantly different with respect to treatment, storage time, or interaction between treatment and time (*p* ≤ 0.05, FDR corrected). Twenty-nine of the 30 proteins showed a gradual increase in concentration during storage in both PI-PAS and C-PAS, indicating some contribution of platelet release.

Hierarchical clustering analysis of the 30 significant proteins with heatmap visualization demonstrated that the proteins can be grouped into three clusters (Figure 2).

Cluster I include four proteins, which all show lower concentrations in PI-PAS compared to C-PAS.. Concetrations were significalny lower at all time points for Eotaxin and on day 2 and 4 for MCP-1, MDC, and IP-10 in PI-PAS compared to C-PAS. In addition, MCP-1, MDC, and IP-10 all display a gradual increase in concentration during storage in both PI-PAS and C-PAS (Figure 2). Cluster II contains eight proteins, TNF-α, sCD40L, GRO, PDGF-AA, PF-4, EGF, IL-7, and sP-selectin, which all showed relatively large increases in concentration during storage in both PI-PAS and C-PAS. In this cluster, the concentration of PF-4 was significantly higher in PI-PAS than in C-PAS on Days 4 (7.8 ± 0.9 μg/mL vs. 5.1 ± 1.5 μg/mL; *p*-value < 0.01) and 7 (13.1 ± 1.4 μg/mL vs. 9.3 ± 2.3 μg/mL; *p*-value < 0.0001). TNF-α concentrations were significantly lower on Day 2 in PI-PAS than in C-PAS (5.1 ± 1.9 pg/mL vs. 9.2 ± 0.7 pg/mL; *p*-value < 0.01) (Figure 3). Cluster III contains 18 proteins; Fractalkine, FGF-2, IL-12P70, TGF-α, GM-CSF, VEGF, IFNγ, IFN-α2, IL-17A, IL-1α, IL-1β, IL-9, IL-10, IL-12P40, IL-15, MIP-1β, MIP-1α, and G-CSF all gradually increased in concentration during the storage period in PI-PAS and C-PAS,. TGF-α, had a significantly lower concentration in PI-PAS than in C-PAS at Day 2 (1.5 ± 0.7 pg/mL vs. 3.4 ± 1.0 pg/mL; *p*-value < 0.05). Three proteins showed significantly higher concentrations in PI-PAS than in C-PAS on Day 7: IL-12P70 (21.3 ± 10.2 pg/mL vs. 12.4 ± 5.3 pg/mL; *p*-value < 0.05); IL-17A (9.6 ± 3.9 pg/mL vs. 5.6 ± 2.5 pg/mL; *p*-value < 0.01); and G-CSF (116 ± 39 pg/mL vs. 81 ± 27 pg/mL; *p*-value < 0.05) (Figure 3). 

IL-13, IL-8, IL-5, IL-1ra, MCP-3, and TNF-β concentrations did not change during the storage time and did not differ between treatments, indicating little or no release of these six proteins from platelets during storage.

## 3. Discussion

There is an increased demand in modern blood banking for high-quality products that are safe for clinical use. Therefore, it is important to evaluate the effects of additional processing technologies that allow longer storage, such as PI treatment, on PC quality, and PSL.

Here, we looked at the concentration of 36 proteins (cytokines, chemokines, and growth factors) in supernatant from stored platelets concentrates. The main goal was to evaluate whether PI using the amotosalen-UVA method influences the concentration and accumulation of these proteins in buffy coat PCs stored under standard conditions over a 7-day period. PCA analysis revealed some differences between PI-PAS and C-PAS, but most of the variation was related to storage time. Four proteins in cluster I (Eotaxin, MCP-1, MDC, and IP-10) gradually increased during storage, but had lower PI-PAS concentrations (Figure 2). This lower concentration in PI-PAS relates to a drop in concentration for these four proteins plus two additional proteins: TNF-α and TGF-α on day 2 after PI treatment (Figure 3). This decrease is a strong indication of treatment-related effects; these six proteins seem to be, to some extent, cleared from the PC during the PI processing. Possible causes are protein degradation caused by UV-light exposure [68] or incubation with the compound absorption device. Our group has shown that specific metabolite species show a significant concentration drop after PI processing, especially after the compound absorption phase [69]. The decrease in levels of specific proteins can be beneficial for patients, especially if these proteins have been implicated in transfusion reactions. At least one of the above-mentioned proteins, TNF-α, has been implicated in febrile nonhemolytic transfusion reactions in transfusion recipients [44,70]. Using Luminex xMAP technology and computerized risk prediction models, it was shown that macrophage-derived chemokine (MDC) poses an increased risk of transfusion-related adverse events at high concentrations [60]. Since the concentration of TNF-α was the same in PI-PAS as in C-PAS by Day 4 or storage, the benefits of a concentration decrease would only apply to fresh (1–2 days old) PCs. For MDC, the concentration gradually increased after Day 2, but was significantly lower throughout the storage period in PI-PAS. 

All the proteins in cluster II are BRM and display a relatively large and significant increase during storage. These proteins are stored in the α-granules of platelets and subjected to de-granulation during activation [3,71,72]. Only one out of the eight proteins in cluster II displayed treatment related effects. PI treatment effected the concentration of PF-4 showing higher levels on Days 4 and 7 (Figure 3). A higher concentration of PF-4 could indicate accelerated degranulation and activation in the PI-PAS. Apelseth et al. reported a higher concentration of PF-4 and TGF-β in PI treated PC compared to untreated control PC on day 5 and 7 of storage. In our analysis we did not observe the same difference regarding TGF-β. These analyses were performed on apheresis PC, using a different storage solution [73]. In a previous, similar in vitro analysis to ours by Johnson et al., there was no statistical difference recorded in the concentration of PF-4 between control an PI treated PC [74]. In the Johnson et al. report there is a slide drop in concentration of PF-4 in the PI treated PC on day two after PI treatment at the same time a steep increase in PF-4 is observed in the untreated control. The largest difference in PF-4 concentration is on day 2 with higher levels in the untreated control. This difference evens out at later points in the storage period with a larger increase in concertation of PF-4 in the PI PC compared to the control. In our analysis we observed a different pattern with equal PF-4 concentrations in PI-PAS vs. C-PAS on day two, and significantly higher concentration on day 4 and 7 in PI-PAS. These different results might relate to sampling and processing of samples, for example in our analysis samples were collected in a 10 mL syringe threw a sterile clave and not using sterile-docking of a sample pouch as in Johnson et al. In addition, in our analysis each unit originated from a 24 donor BC pool compared to eight donors in the Johnson et al. analysis. Three additional proteins, IL-12P70, IL-17A, and G-CSF, assigned to cluster III with more mild increase in concentrations levels during storage show a higher level in the PI-PAS compared to C-PAS. To our knowledge, IL-12P70, IL-17A, and G-CSF have not been seen to be actively stored or secreted by platelets. One hypothetical mechanism of this elevated concentration is the cleavage of proteins that bind to the surface of platelets or accumulation of surface proteins in the PC supernatant due to platelet lysis [71]. In the case of G-CSF, this protein actively binds to platelets, increasing aggregation of activated platelets [75]. This might apply to other proteins that bind to platelet receptors. Proteins with elevated concentration levels during storage with no published evidence of platelet storge or platelet expressed receptors might be a result of degradation of protein complexes or exosomes present in the PC supernatant after collection [71]. The platelet proteome includes proteins specifically packed in to platelets by megakaryocytes as well as endocytosed plasma proteins [76]. Analysis of platelet proteome and secretome show the number of secreted proteins is >300, and a diverse proteome with some proteins represented in the platelet transcriptome [77]. A recent analysis of protein secretion of thrombin activated platelets from 32 healthy individuals detected 894 proteins across all participants and 277 proteins reproducibly detected in all participants with low intensity donor variability [78]. Other analysis of indicate some platelet proteomic donor variation [79].

The PI treatment-related drop in concentration on Day 2 in the PI-PAS needs further investigation, as does the accelerated increase in concentration of some of the proteins after Day 2 of storage. There is some evidence of miRNA regulation of de-granulation. In our previous published data on miRNA in platelets during storage, miRNA-96 was downregulated in platelets treated with PI [80]. miRNA-96 has been implicated in regulation of vesicle-associated membrane protein 8 (VAMP8) that effects the degranulation of platelets [81,82].

Using various markers of platelet activation and granule release, a number of studies have noted accelerated PSL during storage in PI-treated PCs compared to standard PCs [23,73,74]. In our previously published data [80], we also saw some indications of this effect, with increased shedding of the GPIba receptor and surface expression of P-selectin. The molecular basis for increased PSL during storage and the potential effects of additional processing like PI treatment have been extensively studied in recent years, though there have been conflicting results and debate on their clinical relevance. However, despite evidence that the processing steps of amotosalen-UVA treatment affect the biology of stored platelets [73,74,83,84,85,86], numerous clinical trials have demonstrated amotosalen-UVA treatment to be safe and non-inferior to standard platelets in terms of transfusion efficacy. Hemovigilance studies have reported similar or lower rates of adverse events for PI-treated vs. untreated PCs [29,30,73,87,88]. 

## 4. Materials and Methods

### 4.1. Collection, Processing, Storage, and Sampling

Collection, processing, storage, and sampling have been described previously [68]. Briefly, to minimize donor variation effects, a total of 24 buffy coats from healthy ABO-matched blood donors were pooled and split up in two identical smaller pools to obtain double dose PCs by the buffy coat method at the Blood Bank, Landspitali-The National University Hospital of Iceland (Figure 4). This experimental set up included a treatment arm, pathogen inactivation (PI-PAS), and untreated control (C-PAS). PI-PAS received amotosalen-UVA PI treatment (INTERCEPT Blood System, Cerus Europe BV, Amersfoort, Netherlands). PI treatment included the addition of 3 mM amotosalen solution, UVA (340 to 400 nm) light exposure, and incubation with a compound absorption device for 12 to 16 h in a flatbed temperature-controlled incubator (PC900H, Helmer, Noblesville, IN, USA) at 22 ± 2 °C under gentle agitation. All double dose PCs were then split into two single doses, following protocol for patient infusion. The PCs were stored with 35% allogenic donor plasma and 65% platelet additive solution (PAS) in a platelet incubator under standard blood bank storage conditions (22 ± 2 °C with gentle agitation) for seven days. Sampling of PC units was done in a closed sterile system. Baseline sampling was done on Day 1 before PI treatment, with further samples taken on Days 2, 4, and 7 after PI treatment. The experiment was repeated eight times (*n* = 8). 

Media samples for Luminex and ELISA analysis were collected after centrifugation at 1730× *g* at room temperature for 10 min (Sorvall RC5C, Thermo Fisher Scientific, Waltham, MA, USA), separating cells from the spent media. Media samples were stored at −80 °C until analysis. 

The two research arms in this study are referred to herein as follows: (1) non-pathogen inactivated PC in PAS (control), C-PAS; and (2) amotosalen-UVA-treated pathogen inactivated PC in PAS, PI-PAS.

Protein concentrations were analyzed with ELISA and Luminex.

### 4.2. ELISA

Quantikine ELISA (enzyme-linked immunosorbent assay) kits (R&D Systems, Minneapolis, MN, USA) were used to determine the concentrations of two proteins sCD62P/sP-Selectin and CXCL4/platelet factor 4 (PF-4) in PC supernatant, according to the manufacturer’s instructions. PC supernatant (controls and treatments) sampled on Days 1, 2, 4, and 7 were removed from −80 °C storage and thawed on ice in the dark for one hour. ELISA kits and samples were allowed reach room temperature (~22 °C) before use. Sample measurements were performed in duplicate on a microplate reader (Multiskan Spectrum v1.2, Thermo Fisher Scientific, Waltham, MA, USA).

### 4.3. Human Cytokine/Chemokine Magnetic Bead Panel Assay

Undiluted platelet supernatant was analyzed using Luminex xMAP Technology (EMD Millipore Corporation, Billerica, MA, USA) to quantify soluble proteins (growth factors, chemokines and cytokines). The Human Cytokine/Chemokine Magnetic Bead Panel (HCYTOMAG-60K, Millipore) was used; it applies microspheres and fluorescent signaling to quantify 41 pre-selected proteins: EGF, eotaxin, FGF-2, FLT-3L, fractalkine, G-CSF, GM-CSF, IFN-α2, IFNγ, IL-1α, IL-1β, IL-1ra, IL-2, IL-3, IL-4, IL-5, IL-6, IL-7, IL-8, IL-9, IL-10, IL-12P40, IL-12P70, IL-13, IL-15, IL-17A, IL-1RA, IP-10, MCP-1, MCP-3, MDC, MIP-1α, MIP-1β, PDGF-AA, PDGF-AB/BB, RANTES, TGF-α, GRO TNFα, TNFβ, CD40L, and VEGF. Not all 41 proteins were included in the data analysis. RANTES and PDGF AA/BB had very high concentrations and were out of range of the assay, while FLT-3L, IL-2, IL-3, IL-4, and IL-6 were all below the detection limit of the assay. Therefore, the concentrations of 34 proteins were included in the analysis for comparing C-PAS and PI-PAS. Concentration differences were compared on Days 2, 4, and 7. The levels of each protein in the control and treatment groups on Days 2, 4, and 7 were also compared to those in a common baseline Day 1 sample.

### 4.4. Statistical Analysis

Normal distribution of data was assessed analytically with Shapiro–Wilks test and graphicly with quintile–quintile (Q-Q) plots using DATAtab (DATAtab e.U. Graz, Austria). Analysis of variance test ANOVA using GraphPad Prism Version 7.04 (GraphPad Software, Inc., San Diego, CA, USA) was applied to compere normal distributed data and Fridman test using DATAtab for data where a normal distribution was rejected. Differences were considered significant if *p*-values remained below 0.05 after applying the sequential Bonferroni correction method. Hierarchical clustering and heatmap generation were performed using MetaboAnalyst v 3.46. Prior to hierarchical clustering, missing values were estimated using the K-Nearest Neighbors algorithm within the same platform. Hierarchical clustering was performed using Euclidean distance and Ward’s method. Principal components analysis (PCA), a dimensionality reduction technique used to preserve as much of the variance of data as possible in lower-dimensional output [69], was performed using R v3.5.1 (R Development Core Team). 

## 5. Conclusions

In this comparative analysis, storage time is the variable that accounts for most of the change observed in protein concentration. PI treatment does have some effect on protein concentrations during PC storage. Depending on the specific protein and storage duration, increases or decreases in concentration could be seen when comparing non-PI-treated and PI-treated PCs. The effects of these changes on the clinical outcomes of transfusion with PI-treated PCs needs further investigation.

## Figures and Tables

**Figure 1 pathogens-11-00350-f001:**
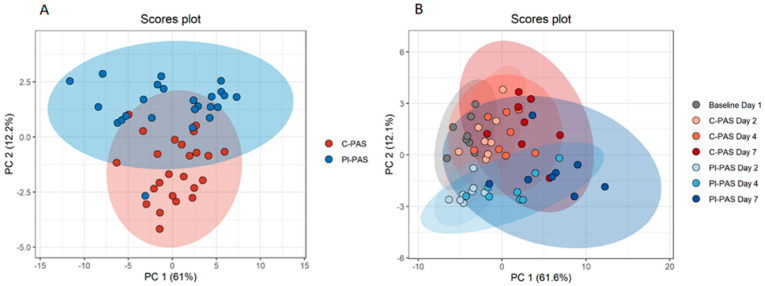
Principal component analysis (PCA) score plots of protein concentrations from pathogen inactivated (PI-PAS) and control (C-PAS) PCs. Measurements were made on Day 1 (baseline, before treatment) and Days 2, 4, and 7 (during storage). (**A**) PCA score plot for Days 2 to 7 for treatment vs. control. Each dot represents collected data for all 36 proteins. Number of dots on plot represents 3 sampling points × 8 biological replicates = 24 dots for each group (**B**) PCA score plot for treatment and sampling day. Each different color dot represents a specific sampling for PI treatment or control. Number of dots represent biological replicates = 8.

**Figure 2 pathogens-11-00350-f002:**
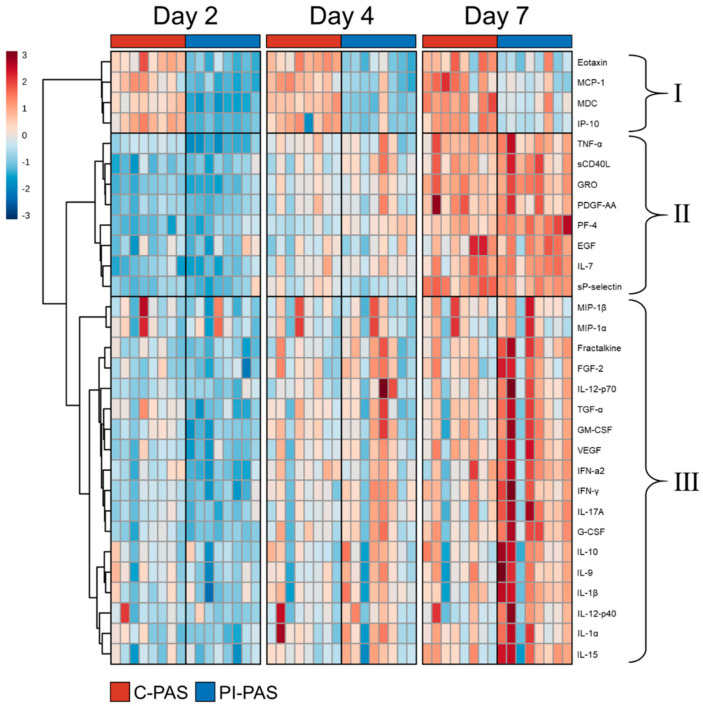
Heatmap of selected proteins measured at Days 2, 4, and 7 of storage. Proteins were selected based on repeated-measures ANOVA significance test (*p* ≤ 0.05, FDR corrected). Hierarchical clustering was performed on auto-scaled normalized abundances using Euclidian distance and Ward’s clustering method. Dark red indicates that the normalized concentration is higher than the average; dark blue indicates that the concentration is lower than average. Three clusters of proteins are labelled with roman numerals.

**Figure 3 pathogens-11-00350-f003:**
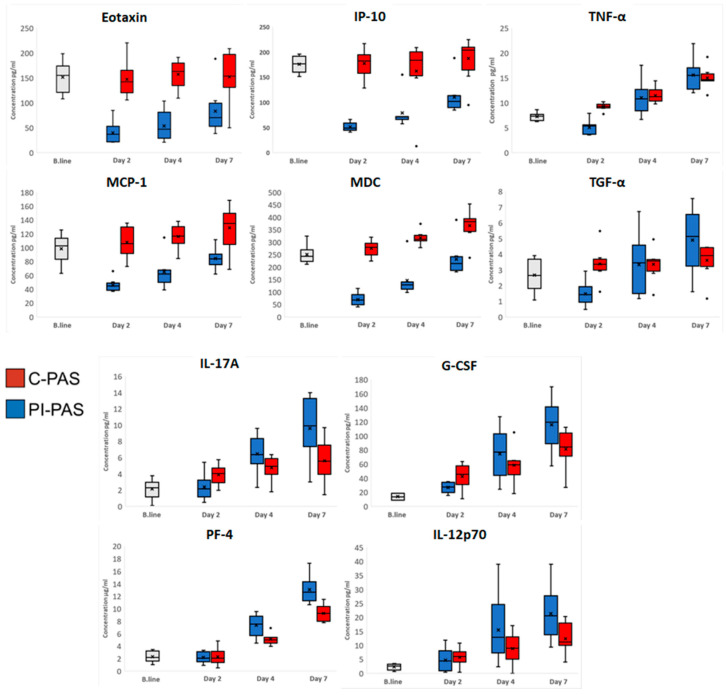
Concentration box plots for proteins identified to be significantly different between treated (PI-PAS) and untreated PCs (C-PAS) using repeated-measures ANOVA and sequential Bonferroni correction. The top and bottom of each box represent the 75th and 25th percentile, respectively, and the central line represents the median. The means are marked with an X, and the outliers with a dot.

**Figure 4 pathogens-11-00350-f004:**
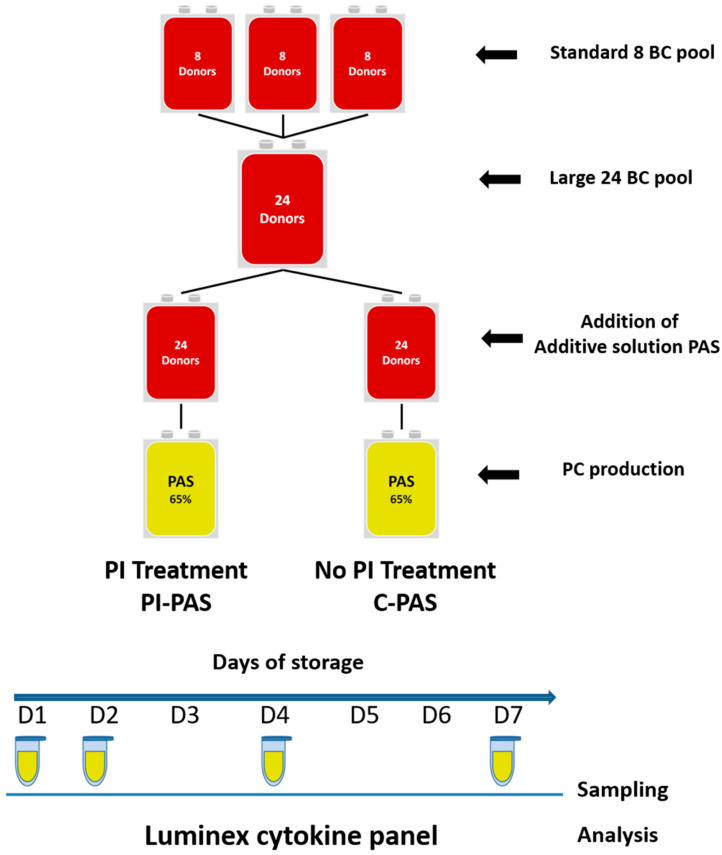
Summary of experimental treatment and control preparation, *n* = 8, adapted from Arnason et al. [68]. Buffy coat (BC), pathogen inactivation (PI), platelet additive solution (PAS).

## Data Availability

Not applicable.

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
