# Peer review of "Protein Concentrations in Stored Pooled Platelet Concentrates Treated with Pathogen Inactivation by Amotosalen Plus Ultraviolet a Illumination"

_pathogens, 2022, doi:10.3390/pathogens11030350_

Round 1
Reviewer 1 Report
Arnason et al. investigated platelet degranulation in response to pathogen inactivation treatment. Overall, manuscript well written and data presented nicely. However, this is a concern regarding the statement author make about the Cluster III protein that "the majority of proteins showing no difference between treatment", which seem snot be the case, at least from looking at heat map. Heat map strongly suggests that there most proteins from Cluster III are indeed present in higher concentrations in PI vs Control. Please provide further details on how the analysis was done?
Also, there are 8 clusters for each protein for each timepoint and each treatment group. What are those? Replicates from the same unit of PCs? Different units of PCs?
Author Response
- Text in manuscript has been edited accordingly to address comment about Cluster III.
- Heat map was generated as visual presentation of all changes in concertation relating to treatment or storage time. Proteins with no change in concentration were excluded . Hierarchical clustering and heatmap generation were performed using MetaboAnalyst v 3.46. Prior to hierarchical clustering, missing values were estimated using the K-Nearest Neighbors algorithm within the same platform. Hierarchical clustering was performed using Euclidean distance and Ward’s method (Included in manuscript)
- In our analysis there where 8 biological replicates (n=8) , each column in the heat map represents an individual unit (replicate), so yes different units of PC
Reviewer 2 Report
In the original paper entitled “Protein concentrations in stored pooled platelet concentrates treated with pathogen inactivation by amotosalen plus ultraviolet A illumination” by dr Arnason et al., the Authors studied the effects of pathogen inactivation on the proteins released in stored blood platelets. In my opinion the paper reports interesting results and is well-written. However, I have one major as well as several minor comments listed below.
Major comment
The Authors used analysis of variances (ANOVA) to compare concentrations of proteins in treated and untreated platelet concentrates. However, using ANOVA is allowed when data has normal distribution. Did the Authors verify normality of distribution? There is a lack of such the information in statistical analysis. Furthermore, if the normal distribution is present, it is rather typical to present data as mean +/- standard deviation not median +/- lower/upper quartile (25%/75% percentile) like it is shown in the Figure 3. Besides, it would be beneficial for the reader to mark significant differences on the Figure 3 (not only in the text of the manuscript).
Minor comments:
Line 24 (abstract): please write in full PSL (platelet storage lesion)
Line 49: method[21] -> method [21]
Line 186: bay Johnson -> by Johnson
Figures 1-4. Please remove the text „Fig 1, 2, 3, 4.” on the top of each figure
Author Response
Mjor comment
- Information on the normal distribution analysis has been added to the manuscript
- We selected to present data in figure 3 with box plots as they show distribution and outliers. Maybe not easy to detect in our figure , the box plot show (X) representing the mean value and a line representing the median value.
Minor comments.
- Text in manuscript has been edited according to minor comments